# Molecular Dynamics Simulation on Solidification Microstructure and Tensile Properties of Cu/SiC Composites

**DOI:** 10.3390/molecules29102230

**Published:** 2024-05-09

**Authors:** Wanjun Yan, Yuhang Lu, Tinghong Gao, Junjie Wang, Xin Tang, Nan Wang

**Affiliations:** 1College of Electronics and Information Engineering, Anshun University, Anshun 561000, China; yanwanjun7817@163.com (W.Y.); 18885719688@163.com (Y.L.); wang_960912@sina.com (J.W.); 2Institute of Advanced Optoelectronic Materials and Technology, College of Big Data and Information Engineering, Guizhou University, Guiyang 550025, China; sweet2505@sina.com (X.T.); 15286692447@163.com (N.W.)

**Keywords:** Cu/SiC composites, molecular dynamics, rapid solidification, mechanical properties

## Abstract

The shape of ceramic particles is one of the factors affecting the properties of metal matrix composites. Exploring the mechanism of ceramic particles affecting the cooling mechanical behavior and microstructure of composites provides a simulation basis for the design of high-performance composites. In this study, molecular dynamics methods are used for investigating the microstructure evolution mechanism in Cu/SiC composites containing SiC particles of different shapes during the rapid solidification process and evaluating the mechanical properties after cooling. The results show that the spherical SiC composites demonstrate the highest degree of local ordering after cooling. The more ordered the formation is of face-centered-cubic and hexagonal-close-packed structures, the better the crystallization is of the final composite and the less the number of stacking faults. Finally, the results of uniaxial tensile in three different directions after solidification showed that the composite containing spherical SiC particles demonstrated the best mechanical properties. The findings of this study provide a reference for understanding the preparation of Cu/SiC composites with different shapes of SiC particles as well as their microstructure and mechanical properties and provide a new idea for the experimental and theoretical research of Cu/SiC metal matrix composites.

## 1. Introduction

In recent years, considerable progress has been made in the field of materials science with the widespread application of new materials, such as synthetic materials [1], metallic glasses [2], and metal ceramics [3], in various engineering and technological fields [4]. Among them, metal matrix composites (MMCs) are widely known for their high thermal conductivity, wear resistance, corrosion resistance, and good thermal stability [5,6,7]. Different reinforcing materials, such as carbon nanotubes, graphene, silicon carbide (SiC) particles, and titanium carbide (TiC) particles, are often used in MMCs to improve their properties [8,9,10,11,12]. As a typical representative of third-generation semiconductor materials, SiC demonstrates excellent thermal, chemical, and mechanical stability [13,14], and it is thus widely used as a reinforcement material for metal substrates such as Al and Cu [15,16,17]. Owing to their excellent mechanical properties, high wear resistance, and thermal conductivity [18], Cu/SiC composites are considered ideal “structure-function integrated” composites [19] and have thus been widely used as engineering materials for various conditions requiring high thermal conductivity and excellent wear resistance [20,21].

Cu/SiC composites have gained widespread research attention, with many researchers comprehensively studying the preparation, mechanical properties, and thermal properties of this material. Sumathi M et al. [22] performed cold upsetting experiments on sintered Cu-SiC preforms to evaluate their deformation characteristics. Schubert et al. [23] prepared Cu/SiC composites using powder metallurgy. Furthermore, improvements were achieved in the bonding strength and thermophysical properties of the composites through a vapor-deposited molybdenum coating on SiC powders to control the detrimental interfacial reactions. Li et al. [24] used molecular dynamics (MD) simulations to investigate the strengthening mechanism of SiC particles during the edge dislocation motion of Cu–matrix nanocomposites under shear loading. Xiong et al. [25] revealed the generation of thermal residual stresses, dislocations, and incomplete stacking fault facets during cooling in the case of ideal Cu/SiC composites; they determined that the combined effect of the stresses induced by the mismatch in the coefficient of thermal expansion (CTE) and the thermodynamic state of the metal resulted in the rapid generation of dislocations. Furthermore, nanosized SiC-reinforced Cu–matrix nanocomposites were manufactured, pressed, and sintered at 775 °C and 875 °C in an argon atmosphere [26]. X-ray diffraction and scanning electron microscopy were performed to characterize the microstructural evolution. Moreover, their density, thermal expansion, mechanical, and electrical properties were analyzed.

Compared with Cu/SiC composites, Cu/graphene composites have also been widely studied. Recently, a novel 3-dimethyl (3D) graphene origami-reinforced metal nanocomposite has been proposed and designed [27]. It is found that the intrinsic brittleness of Gr can be alleviated by transforming its original 2D shape into a 3D origami shape. Ho et al. [28] reported that Miura-GOri is super-flexible with a negative Poisson’s ratio (NPR) even under large tensile and compressive strains. In addition, Zhang et al. [29] have experimentally measured the strength enhancement of copper composites reinforced by Y-shape graphene. The interpenetrating structural feature of the as-obtained composites promotes the interfacial shear stress to a high level and thus results in significantly enhanced load transfer strengthening and crack-bridging toughening simultaneously. Regarding copper/sic composites and copper/graphene composites, both have their unique excellent properties. Therefore, the research on the two kinds of composite materials has also attracted much attention.

Rapid solidification technology has been extensively studied as a material preparation technique. Gao et al. [30] investigated the formation mechanism of dislocations and twin structures of Ti_3_Al during rapid solidification using MD. According to the synthesis and decomposition of different types of dislocations, the formation and evolution of crystal defects under different models were analyzed. In addition, some researchers used MD simulations to study the rapid solidification process of a multiprimary-element CoFeNiPd alloy when considered as a prototypical single-phase high-entropy alloy (HEA) [31]. The results of these simulations suggest that rapid solidification accompanied with heterogeneous nucleation could be a promising approach for controlling the microstructure and properties of HEAs. Colin et al. [32] investigated the impact of incorporating NiAl intermetallic compound Cu on the microstructure and lattice parameters of the β-Ni(Al,Cu) and γ′-(Ni,Cu)_3_Al phase. As-cast and rapidly solidified specimens were characterized using X-ray diffractometry, scanning, and transmission electron microscopy techniques. Papanikolaou et al. [33] used MD simulations to investigate the solidification process of pure Al and the effect of the cooling rate on the final properties of the solidified material. Numerous Al atoms were used to investigate the grain growth over time and the formation of stacking faults during solidification. Lee et al. [34] explored the effect of the cooling rate on the crystallinity and mechanical properties of thermoplastic composites. The results show that, in the crystallinity range studied, the tensile properties are not affected significantly by changes in crystallinity, while the fracture toughness depends strongly on the crystallinity.

Many comprehensive studies have also been conducted on the mechanical properties of Cu/SiC composites. For example, Akbarpour et al. [20] prepared Cu/SiC nanocomposites using mechanical milling and hot-pressing processes. The results showed that the hybrid nanocomposite demonstrated higher wear resistance, a lower friction coefficient, and enhanced compressive strength than those of the microcomposite. Yang et al. [35] investigated the mechanical behavior of co-continuous Cu/SiC nanocomposites using MD simulations. The Young’s modulus predicted through the MD simulation satisfactorily conformed with the results of the micromechanics methods. For other materials, recent studies have comprehensively studied the mechanical properties of Al_2_O_3_, focusing on fracture toughness, surface energy, Young’s modulus, and crack propagation [36]. Further, interfacial properties played an important role in the plastic deformation of the nanocomposites. Fathalian et al. [37] used the density functional theory (DFT) calculation method to study the mechanical and electronic properties of Al (111)/6H SiC composites. The results show that C and Si vacancies reduce interfacial adhesion near the interface, while Al vacancies have little effect. Tahani et al. [38] use the molecular dynamics method to study the interdiffusion at the interface of α-Al_2_O_3_/AlSi12. The results demonstrate that the thickness of the interdiffusion zone is proportional to the annealing temperature and time, and Al- and O-terminated interfaces exhibit similar interdiffusion properties. At the same time, the mutual diffusion of the SiC/Al interface is also simulated [39]. It is found that the mutual diffusion coefficient increases with the increase in temperature and annealing time.

The effect of particle shape affects many other hybrid materials, including polymers. For MD simulations, Jabbarzadeh et al. [40] explored the origin of enhanced and delayed crystallization in nanocomposite polymers. It is shown that although crystallinity is affected by the size of the nanoparticles and their volume fraction, their combined effect can only be measured by the interparticle free space and characteristic size of the crystal. In addition, the effect of the size, shape, and volume fraction of the nanoparticles on the crystallization of the nanocomposite polymer was explained [41]. It is shown that decreasing the size of the particles at the same volume fraction results in a lower final crystallinity regardless of the shape. Similarly, for the same particle size, increasing the volume fraction results in a lower crystal growth rate and final crystallinity. Experimentally, Kumar et al. [42] discuss theoretical advances in the field of polymer nanocomposites. We focus on open questions in the field and build around five separate themes. Papananou et al. [43] tuned the crystallinity of the polymer by the appropriate selection of inorganic nanoadditives. The tuning of the degree of crystallinity of a semi-crystalline polymer is achieved in polymer nanocomposites by utilizing nanoparticles of different sizes as well as their mixtures in ternary systems.

In addition, researchers have recently investigated related materials through ab initio MD (AIMD). Sangiovanni et al. [44] used density-functional MD—accounting for van der Waals interactions—to identify the reaction pathways resulting in the dissociation of the trimethylindium (TMIn) precursor in the gas phase as well as on top- and zero-layer graphene. The simulations revealed how collisions with hydrogen molecules, intramolecular or surface-mediated proton transfer, and direct TMIn/graphene reactions facilitate TMIn transformations, ultimately enabling the delivery of In monomers or InH and CH_3_In admolecules on graphene. Lundgren et al. [45] employed AIMD simulations to assess the thermal stability and mechanical properties, such as the elastic modulus and stress–strain curves, of two-dimensional (2D) InBi; they discovered that 2D InBi and its heterostructures with graphene exhibit high thermal stability, elasticity, and resistance to fracture.

Despite previous advances and developments in the study of Cu/SiC composites, these composites have rarely been analyzed for high-temperature cooling. In the study of the cooling process of Cu/SiC composites by Xiong et al. [25], possible factors affecting the mechanical properties of metal matrix composites were mentioned, including the volume fraction of the reinforcement, the particle size and distribution, and the fact that a ceramic particle can be made of any shape. Therefore, based on this, this paper investigates the effect of different shapes of SiC particles on composites. The changes in radial distribution functions, displacement vectors, structures, and dislocations of Cu/SiC composites during rapid solidification were investigated in detail using molecular dynamics methods. Furthermore, in this study, cooled Cu/SiC composites were subjected to tensile treatment, and stress–strain curves and stress–distribution diagrams were plotted to evaluate the mechanical properties of these composites.

Figure 1 shows a schematic of the ideal model of Cu/SiC composites. As shown in Figure 1, the model has a side length of 86.64 Å. Three shapes of SiC particles, i.e., S1 (sphere), S2 (cylindrical), and S3 (cubic), were introduced into the Cu atoms (Figure 1b–d).

## 2. Results and Discussion

### 2.1. RDF

The RDF [*g*(*r*)] is a commonly used function to gain insight into atom interactions between two similar or different species and can be used to describe the structural behaviors of a system, mainly for liquid and amorphous materials [46]. Figure 2 shows the total RDF plots of the composites at 200 K for different models. The three insets in the figure show the structural distribution of the three models rapidly solidified to 200 K at a cooling rate of 10^11^ K·s^−1^.The green, red, and blue atoms represent the FCC, HCP, and body-centered-cubic (BCC) structures, respectively, and the gray atoms represent other structures.

Figure 2 shows that the S1 model displays the highest main peak, followed by the S3 and S2 models, implying that the highest degree of localized ordering is found in the spherical SiC composites, whereas the lowest degree is found in the cylindrical and orthorhombic SiC composites. In addition, compared with the other two models, the S1 model possesses the fewest number of amorphous atoms and the highest degree of crystallization.

### 2.2. Atomic Displacement Changes

Figure 3 shows the displacement vector plots for the three samples cooled to 200 K. Because SiC always remains in a steady state, the displacements mainly occur in the Cu atoms. As shown in the figure, the atomic displacements in the composite material are in a disordered state, with a small number of Cu atoms displaying considerably small displacements, while some display considerably strong displacements. As shown in Figure 3b, near the edge of the SiC particles, the disorder of the atoms is amplified, with numerous atoms moving in all directions. In addition, Cu atom displacements are observed to diffuse toward the SiC particles. This could be attributed to the fact that the interaction between the high temperature and interfaces results in the diffusion of the Cu atoms into the stationary SiC particles during the cooling process. Thus, the direction of the displacements appears disordered [25]. In addition, the three shapes of SiC particles resulted in considerable differences at the interfaces between the SiC particles and Cu atoms; however, the differences in the interfaces did not lead to substantial changes in the direction and magnitude of the displacements within the composites. Figure 3 clearly depicts the displacement variations within the composite; the different interfaces produced by the different shapes of SiC did not considerably affect the atomic displacements.

### 2.3. Distribution of FCC and HCP Structures

Different material microstructures result in different macroscopic properties [30]. The modeling of the composites with different SiC shapes resulted in different structural changes during the rapid solidification process of Cu/SiC composites. Figure 4 presents the two structures that are the most abundant and account for the largest portion of the cooling process, i.e., the FCC and HCP structures. Since SiC particle structural properties have been kept in a stable state, the structural changes in the Cu atoms are mainly referred to here.

As shown in Figure 4, the S1 model shows the largest percentage of FCC structures at 200 K, accounting for 73% of the total atom ratio and the smallest percentage of HCP structures at just 11%. Conversely, the S2 model displays the smallest and largest percentages of FCC and HCP structures at 30% and 48%, respectively. The S3 model structure exhibits intermediate percentages of 53% and 26%, respectively. Among the three models, the S2 model displayed a higher percentage of cooled HCP structures than FCC structures. The inset shows the structural evolution curves during this phase of the rapid solidification process in the temperature range of 700 K < T < 1100 K. The structure evolution curves are shown in the inset. Figure 4 shows that the S3 model preferentially starts to crystallize at a temperature of 1035 K, and the S1 model starts to crystallize rapidly at approximately 960 K. Furthermore, the S2 model produces a small portion of crystals at approximately 1020 K before it starts to crystallize in a large area until approximately 810 K. The S3 model displays the smoothest crystallization process among the three models. These results show that different SiC shapes not only affect the structural distribution of the composites but also lead to changes in the temperature nodes of the composites from the amorphous to crystalline transformation. Among the three models, the S1 model exhibits the best crystallization effect, which may be attributed to the relatively regular interface between the spherical SiC particles and Cu atoms.

Figure 5 shows the structural evolution of the crystallization process of the three samples. CNA analysis was used to preserve the FCC and HCP structures, allowing for a clearer observation of the crystallization process of the models and the resulting stacking faults. As shown, in terms of the initial crystallization, models S1 and S3 started to crystallize from the top with aggregation into clumps. Subsequently, the FCC and HCP structures continued to be generated and aggregated above and below the models. S1 crystallizes rapidly and in an orderly fashion between 958 and 954 K, where the FCC and HCP structures can be observed to be ordered into clusters; in addition, the formation of many stacking faults is initially observed, as shown in Figure 5(a3). On cooling to 900 K, most of the amorphous solid is converted to crystal, with S1 comprising a small number of stacking faults. The S3 sample was slightly disorganized during the rapid crystallization process between 1030 and 1025 K, with the FCC and HCP structures divided into multiple blocks, as shown in Figure 5(c3). A higher number of stacking faults were observed to be distributed throughout when the sample was cooled to 800 K (Figure 5(c4)). As shown in Figure 5(b1–b4), model S2 showed the most peculiar observation; it started crystallizing from 1020 K. At 1000 K, the disordered distribution of FCC and HCP structures within the sample could be observed clearly. When the sample was cooled to 810 K, more HCP structures were observed than FCC structures. Finally, a large number of stacking faults were distributed within the system at 700 K. In summary, the more ordered the sample is in the production of FCC and BCC structures during the rapid solidification process, the better the final crystallization and the fewer the stacking faults.

### 2.4. Distribution and Evolution of Dislocations

The samples generated numerous dislocations during the rapid solidification process (Figure 6). Figure 6 shows the DXA snapshots and dislocation distributions of the samples after cooling to 200 K as well as changes in the dislocation density during the cooling process. The dislocation density is calculated as the total dislocation length per unit volume [47]. Figure 6a shows that dislocations appear on the surface of the sample, with a small number of other atoms distributed around the dislocations (in the black dashed box). After removing the atoms, these ranges (in the black dashed box) are observed to be the main distribution points of dislocations within the model (black dashed box in Figure 6b). Although all the dislocations were concentrated around the SiC particles, most of the dislocation lines extended outward from the Cu–SiC interface. As shown in Figure 6b, sample S1 displays fewer dislocation lines with a more scattered distribution. Conversely, numerous dislocation lines were observed for samples S2 and S3, with a more concentrated distribution. This could be attributed to the different shapes of SiC particles resulting in different interfaces between the Cu and SiC particles. In addition, dislocations were not generated within the SiC particles during the cooling process and were generated only in the metal. This result has been previously verified for MMCs [4,48,49]. Figure 6c shows the dislocation density variation curves for the three samples, wherein the dislocation density of S1 increases sharply from 960 K and then decreases rapidly, finally remaining steady. The dislocation densities of S2 and S3 gradually increased from 810 and 1030 K, respectively. In addition, S2 produced large fluctuations in the cooling phase from 500 to 200 K, while S3 gradually reached a smooth phase after 600 K. Ultimately, the S1 and S2 samples demonstrated the lowest and highest dislocation densities, respectively. This is because sample S1 produced the least amount of stacking faults and the best crystallization. Thus, different SiC shapes considerably influence the dislocations produced via the rapid solidification of the Cu/SiC composites.

Figure 7 shows the dislocation evolution of the samples during the rapid increase in dislocation density during the rapid solidification process. The figure demonstrates that at a temperature of 950 K, a substantial number of dislocation lines are observed in the S1 sample; a few dislocations are present in the S3 sample, while no dislocations are generated in the S2 sample. As the temperature decreases, the dislocation count in the S1 sample rapidly decreases and gradually stabilizes, while the dislocation count in the S3 sample gradually increases, concentrating within a certain range at 600 K. The S2 sample starts to show dislocation lines gradually only at a temperature of 800 K, followed by a gradual increase, and the distribution of dislocation lines at 600 K is also more concentrated. These phenomena are consistent with the changes in dislocation density in Figure 6c.

Figure 6 and Figure 7 show that the S2 and S3 dislocations are mainly clustered in a certain interval; therefore, the dislocation evolution was sliced and analyzed within that interval. The top of S2 and the bottom of S3 dislocations were analyzed, as shown in Figure 8. For clarity, Figure 8(a1,b1) show the snapshots of DXA slices, with FCC and HCP structures preserved, and Figure 8(a2,b2) show the snapshots of slices with dislocation evolution. During the cooling process, a large number of ordered HCP structures appeared around the dislocations, and the generation of dislocations was accompanied by the appearance of stacking faults, as shown in Figure 8(a1,b1). As the dislocations in S2 are more ordered, the stacking faults were also observed to be ordered (red atoms in Figure 8(a1)). In contrast, the dislocation distribution in S3 was more disorganized, and thus the stacking faults in the slices showed different orientations and shapes (Figure 8(b1)). In addition, dislocations in both the model slices increased with decreasing temperature. As shown, the dislocation lines continuously interact with each other during the growth process and evolve under both aggregation and separation variations. As shown in Figure 8(a2), numerous dislocations in the upper part intersected to form a dislocation node at 200 K. However, the dislocation lines were shorter at 400 K, and thus this phenomenon could be attributed to the extension and interaction. In S3, two neighboring dislocation nodes were also generated at 600 K; however, after dislocation interactions, only one dislocation node remained, while the others dissipated by the time the sample was cooled down to 400 K. The results showed that dislocation generation and stacking faults are closely related. When many dislocations are clustered in a certain region, the interaction between the dislocations generates dislocation nodes and simultaneously results in the dissipation of the already generated dislocation nodes.

### 2.5. Uniaxial Tensile in Three Different Directions Properties at 200 K

Stresses and strains are measures used to describe the internal forces and the resulting deformations within materials [50]. Jami et al. [51] explored the effect of particle shape on mechanics of impact in the deposition of titanium nanoparticles on a titanium substrate. They also calculated the time-averaged stress and strain of the particles and plotted the stress–strain curve [52]. Accordingly, we calculated and plotted the stress–strain curves of the composites during the tensile process with different SiC particle shapes. The cooled samples were subjected to uniaxial tensile in three different directions with the NVT ensemble and a strain rate of 1 × 10^9^ s^−1^, and the Young’s modulus was calculated, as shown in Figure 9. The inset shows CAN snapshots at a strain of 0.2, at which point the models were observed to differ in terms of their structure after uniaxial tensile in three different directions. Here, the S2 sample showed the greatest difference, with more FCC atoms on the surface of the samples after *Z*-axis stretching; further, fewer stacking faults were clearly observed than those in the *X*- and *Y*-axis-stretched samples. This is due to the relatively homogeneous interface between the cylindrical SiC particles and Cu atoms in S2 in the *Z*-axis direction (Figure 1c). For *X*-axis stretching, the tensile strength of S1 is the largest and that of S3 is the smallest; for *Y*-axis stretching, the tensile strength of S2 is the largest and that of S3 is the smallest; and for *Z*-axis stretching, the tensile strength of S1 is the largest and that of S3 is the smallest. Furthermore, S1 demonstrated the largest Young’s modulus for uniaxial tensile in three different directions, with corresponding Young’s modulus values of 172.129, 178.671, and 188.916 GPa in the X-, Y-, and Z-axes directions, respectively. In addition, S2 displayed the smallest Young’s modulus for *X*-axis stretching, with 142.944 GPa. Conversely, S3 showed the smallest Young’s modulus values for Y- and Z-axes stretching, with 166.124 and 163.410 GPa, respectively. These results showed that the cooled S1 and S3 samples displayed the best and worst mechanical properties, respectively.

Figure 10 shows the evolution of the deformation of the three samples under *X*-axis stretching for different strain values. As shown in the figure, local strains develop with the progress in stretching, thus forming shear bands. It can be found in the figure that the central SiC particle part does not produce local strain, so we can infer that SiC remains in a stable state in the case of stretching as in the cooling process, and no structural changes occur. Unlike S1 and S2, a large amount of localized strain generation is observed at the interface of Cu and SiC in the S3 model for the strain of 0.08, where significant deformations can be observed. That is, the S3 model was the first to start deforming during the stretching process, and this result is consistent with the description of the stress–strain curve in Figure 9a, where S3 is the first to reach the tensile strength point. When the strain reached 0.095, a large amount of localized strain was observed at all sample interfaces with significant deformation. This suggests that the Cu and SiC interface is responsible for the initial deformation of the samples. Ultimately, S3 underwent the greatest degree of deformation among the three samples owing to the presence of the greatest amount of localized strain and the lowest tensile strength among the three samples (Figure 9a). These results have been similarly concluded by Doan et al. [53]. Model S1 displayed the smallest degree of final deformation, with the least localized stress and the largest tensile strength among the three samples; this could be attributed to the relatively homogeneous interface between Cu and sphere SiC particles in S1. The results showed that the interface is responsible for the initial deformation of the samples, and a relatively homogeneous interface improves the tensile strength of the system.

## 3. Simulation Method

### 3.1. Geometrical Model

It is well known that 4H-SiC is considered an attractive material for next-generation semiconductor power devices [54], and its excellent performance and high precision requirements have led to an increasing number of scholars to study it in the last two years. 4H-SiC has unique physical and mechanical properties that give it excellent stability in high-temperature environments [55,56]. In addition, 4H-SiC has high thermal conductivity and chemical stability, making it an ideal matrix for composites [57]. In this study, the condensation and tensile properties of Cu/SiC composites centered on 4H-SiC particles of different shapes were investigated using MD simulations. Figure 1 shows a schematic of the ideal model of Cu/SiC composites. The model is based on some previous composite studies [24,25,58], which construct a composite of Cu-wrapped SiC particles. As shown in Figure 1, the model has a side length of 86.64 Å. Three shapes of SiC particles, i.e., S1 (sphere), S2 (cylindrical), and S3 (cubic), were introduced into the Cu atoms (Figure 1(b–d)). The SiC particles were immobilized at the center of the Cu atoms with a volume fraction of ~8.2%. The volume fraction of SiC particles is calculated from the ratio between the number of SiC atoms and the total number of atoms contained in the composite. The rebar was controlled to maintain the same volume fraction by adjusting the size of the SiC particles of the three shapes. For the volume fraction, a too-low SiC volume fraction may not achieve the desired purpose, resulting in the observation that the effect of the three SiC particle shapes is not clearly distinguishable. On the other hand, due to the consideration of experiments as well as practical applications, a too-high SiC volume fraction may lead to composite materials that are difficult to shape or process. Therefore, after comprehensive consideration, we chose SiC particles with an 8.2% volume fraction. 4H-SiC is a hexagonal crystal system with crystal orientation [0001] [59]. Each composite model contained a 4H-SiC particle in the center of a cubic Cu matrix that oriented along the principal [100], [010], and [001] axes, respectively [25]. Each model comprised ~55,000 atoms and used periodic boundary conditions in the X, Y, and Z directions with a time step of 0.001 ps and energy minimization based on the conjugate gradient method. Setting the simulation box edge length to an integer multiple of the copper lattice constant (3.61 Å) [60], 86.64 Å, was conducted to eliminate the influence of boundaries on the model and to ensure the periodicity of the simulated system. The lattice constant describes the arrangement of atoms in a crystal, and integer multiples of the edge lengths ensure the integrity and consistency of the lattice arrangement.

The aforementioned model was subjected to solidification simulations in an isothermal–isobaric (NPT) ensemble. The models were initially heated from room temperature to 1600 K at a rapid heating rate of 10^14^ K·s^−1^, surpassing the melting point of Cu. The melting point of copper has been studied in numerous molecular dynamics simulations [61,62,63] and does not differ much from the experimental melting point. The melting point of copper obtained from the simulated potential energy–temperature curve in this study is 1373 K [64,65,66], which is basically the same as the experimental melting point of 1356 K [67,68]. Next, equilibration was conducted at 1600 K for 60 ps, after which the model was cooled to 200 K at a cooling rate of 10^11^ K·s^−1^. Notably, as the temperature of 1600 K was considerably lower than the melting point of SiC, i.e., 3250 K [69], the structural properties of SiC were relatively stable during this period. After solidification, the model was further subjected to 100 ps NPT relaxation, and the overall microstructure and thermodynamic properties of the system were found to be stable. Next, the model was subjected to uniaxial tensile in three different directions simulations using a canonical (NVT) ensemble; the strain rate was 1 × 10^9^ s^−1^. By fixing the remaining directions, stress is applied to the single axis of the composite, the X, Y, or Z axis, so that the composite is deformed in that direction. The uniaxial stretching process begins with a 100 ps relaxation of the model to ensure that the system is in equilibrium. Then, keeping the other two directions unchanged, stretching is performed in a fixed direction. The NVT ensemble is simulated by keeping the volume and temperature constant, and uniaxial stretching can represent the mechanical properties of the material well under the condition of constant volume [70,71].

The solidification of liquid metal can introduce various defects such as dislocations, stacking faults, and twins [72]. Observing the evolution of these defects is a key objective of this study. In addition, this temperature is far below the melting point of SiC and does not result in any changes in SiC. This study was focused on investigating the effect of different SiC shapes on the composites; therefore, ensuring that the SiC particles did not undergo any structural changes was essential. The system was cooled to 200 K because this temperature ensures the complete solidification of the composite material, allowing for a more clear observation of the solidified state and microstructure. This phenomenon has been reported by numerous studies [30,46].

The proposed MD simulation process was performed using a large-scale atomic/molecular massively parallel simulator (LAMMPS) [73]. The atomic structure of the material was visualized using the OVITO software [74], wherein the common neighborhood analysis [75] was used to analyze the microstructural changes in the material; in addition, the dislocation extraction algorithm (DXA) [76] was used to observe the dislocation evolution during material stretching.

### 3.2. Interatomic Potentials

In MD simulations, the potential function is an important parameter for describing interatomic interactions. The accuracy of the simulation results is closely related to the selection of the correct potential function. Therefore, potential functions that correctly describe the interatomic interactions between Cu and SiC must be carefully selected. For determining the interaction potential between the Cu atoms, the embedded-atom method-derived potential [77] was selected. This potential has been successfully applied at various high temperatures and pressures to demonstrate its accuracy in terms of temperature-dependent mechanical properties [78,79,80]. In addition, the potential can accurately characterize dislocation nucleation, multiplication, and stacking fault formation in Cu under multiple conditions including uniaxial and shear loading [81].

To determine the interaction potential between SiC atoms, we selected the classical Tersoff potential [82]. Since its development, this potential function has been widely used for simulating SiC materials and has been able to exhibit good accuracy in various applications [83,84,85], with the advantages of reduced time and small computational effort [48]. In addition, the Morse potential was used to characterize the interaction between Cu and SiC, with the energy being expressed as $E=D_{0}\left[e^{-2\alpha \left(r-r_{0}\right)}-2e^{-\alpha \left(r-r_{0}\right)}\right]$, where $D_{0}$ represents the cohesion energy, α represents a constant parameter, r represents the distance between the two atoms, and $r_{0}$ represents the distance between atoms in the equilibrium state [25]. Parameters $D_{0}$, α, and $r_{0}$ for the Cu–C interaction were obtained as 0.087 eV, 5.14 Å^−1^, and 2.05 Å, respectively [24]. For Cu–Si, $D_{0}$, α, and $r_{0}$ were obtained as 0.435 eV, 4.6487 Å^−1^, and 1.9475 Å, respectively [24]. The cut-off radius is 2.5 Å.

## 4. Conclusions

In this study, MD simulations were conducted to investigate the rapid solidification process of Cu/SiC composites containing 4H-SiC particles of different shapes from 1600 to 200 K at a cooling rate of 10^11^ K·s^−1^. The initial crystal orientations of the Cu matrix along the x, y, and z axes were [1 0 0], [0 1 0], and [0 0 1], respectively. The microstructural evolution of three composite models during cooling and the tensile mechanical properties of the cooled models were analyzed for a fixed simulation box edge length under periodic boundary conditions. This study provides new ideas for the experimental and theoretical study of Cu/SiC MMCs as well as a theoretical reference for the preparation of such composites. The following conclusions can be drawn from the simulation results.
Based on the RDF results, the comparison with the other two samples demonstrated that S1 exhibits the highest degree of local ordering and the best crystallization.The interaction between a high temperature of 1600 K and the interface resulted in disordered atomic displacement in the Cu/SiC composites. However, the interfacial differences caused by different shapes of SiC particles did not considerably affect the displacements within the composites.During the cooling process, the initial crystallization temperatures of the SiC samples with different shapes differed, where S3 was observed to crystallize first. In addition, the final crystallization effect of the samples was related to the degree of ordering of the FCC and HCP structures produced via the cooling process. The more ordered the formation is of the two structures, the better the final crystallization effect is and the lesser the stacking faults.The analysis of the microstructures in the simulations shows that the difference between the Cu and SiC interfaces within the model has a significant effect on the generation of dislocations and that the generation of dislocations in the Cu atoms is closely related to stacking faults. Among the three samples, S1 displayed the lowest dislocation density, while S2 and S3 displayed high dislocation densities. The generated dislocations were mainly distributed in a region around the interface, where dislocation interactions generate dislocation nodes and also make the already generated dislocation nodes disappear.The uniaxial tensile in three different directions results showed the S1 and S3 samples to possess the best and worst mechanical properties, respectively. Tensile stretching leads to a lot of local strains inside the model and the formation of shear bands at the interface, where the Cu and SiC interface is responsible for the initial deformation of the sample. Ultimately, differences in Cu and SiC interfaces caused varying degrees of deformation, and uniform Cu and SiC interfaces could increase the tensile strength of the system.

In future work, the effects of temperature and SiC particle size on the Cu/SiC composites during solidification and the changes in the mechanical properties of the composites will be further investigated in order to prepare ideal Cu/SiC composites with excellent properties.

## Figures and Tables

**Figure 1 molecules-29-02230-f001:**
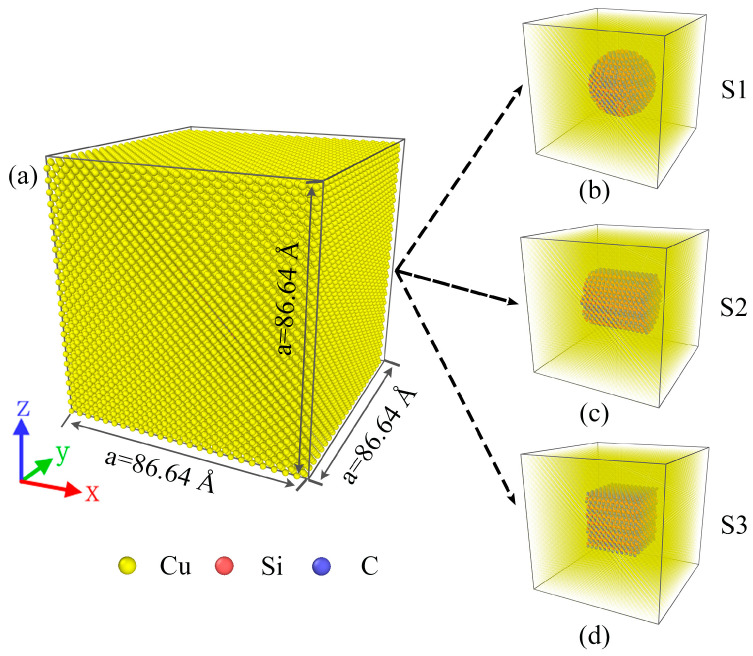
(**a**) Model diagrams of Cu/SiC composites, three models after copper atom transparency: (**b**) S1, (**c**) S2, and (**d**) S3.

**Figure 2 molecules-29-02230-f002:**
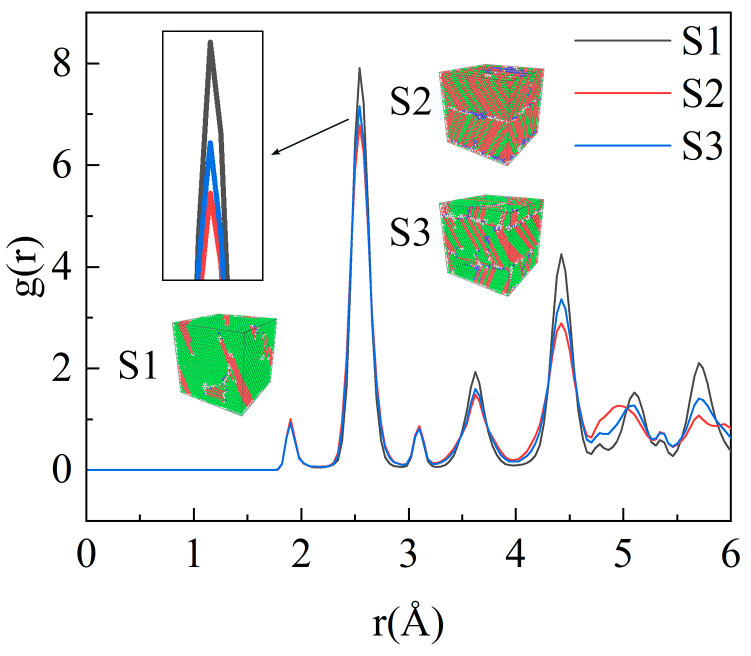
Radial distribution function [g(r)] of the Cu/SiC composites cooled to 200 K.

**Figure 3 molecules-29-02230-f003:**
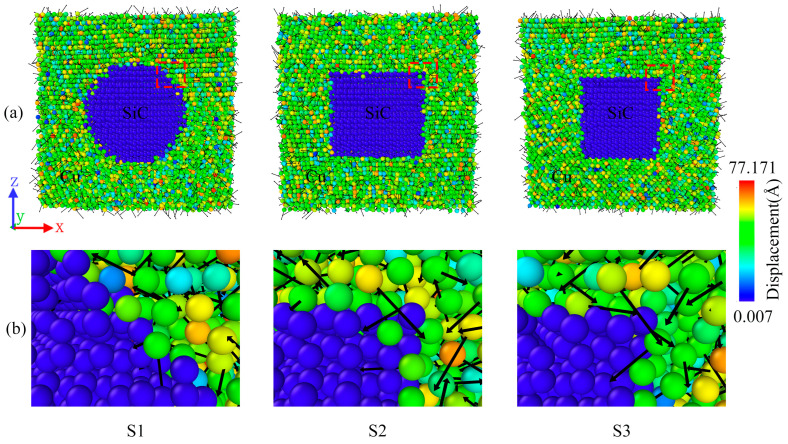
(**a**) The slice shows the change in the displacement of atoms in the three composite models. (**b**) is a magnified view of the interface in the red dashed box in (**a**). The black arrow pointing represents the direction of displacement, and the length represents the size of the displacement. Meanwhile, the closer the color is to red, the stronger the displacement is.

**Figure 4 molecules-29-02230-f004:**
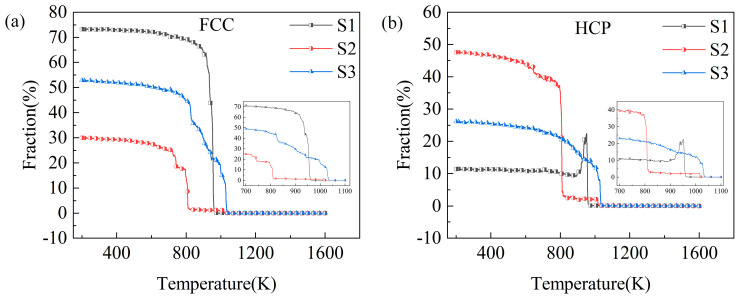
Volume fractions of FCC and HCP structures in different models at a cooling rate of 10^11^ K·s ^−1^. (**a**) FCC structures, (**b**) HCP structures.

**Figure 5 molecules-29-02230-f005:**
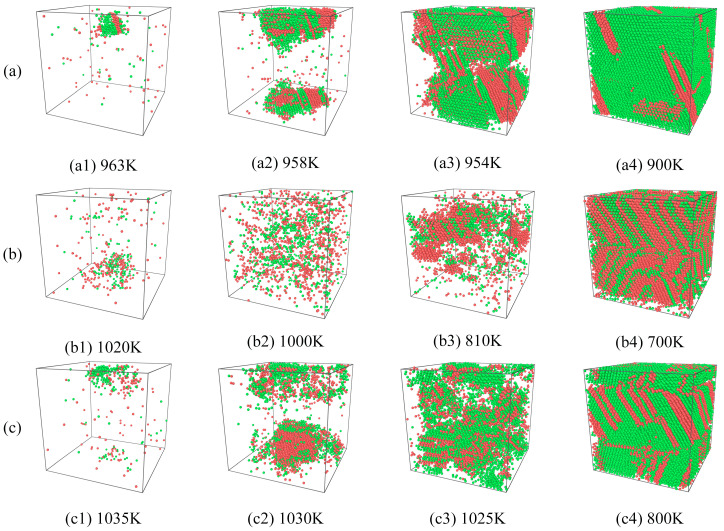
The structural evolution of FCC and HCP in different models. (**a**) S1, (**b**) S2, (**c**) S3. The green atoms are FCC structures, and the red atoms are HCP structures.

**Figure 6 molecules-29-02230-f006:**
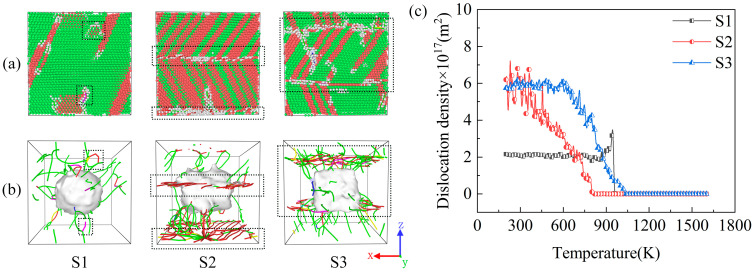
(**a**) A DXA snapshot of the model at 200 K, (**b**) the dislocation distribution of the model at 200 K, and (**c**) dislocation density variation. Green represents FCC atoms, red represents HCP atoms, and gray represents other atoms. The green lines represent Shockley partial dislocations (Burgers vector 1/6 < 1 1 2 >); the blue lines represent perfect dislocations (Burgers vector 1/2 < 1 1 0 >); the magenta lines represent stair-rod dislocations (Burgers vector 1/6 < 1 1 0 >); the yellow lines represent Hirth dislocations (Burgers vector 1/3 < 1 0 0 >); the light blue lines represent Frank dislocations (Burgers vector 1/3 < 1 1 1 >); the red lines represent other dislocations.

**Figure 7 molecules-29-02230-f007:**
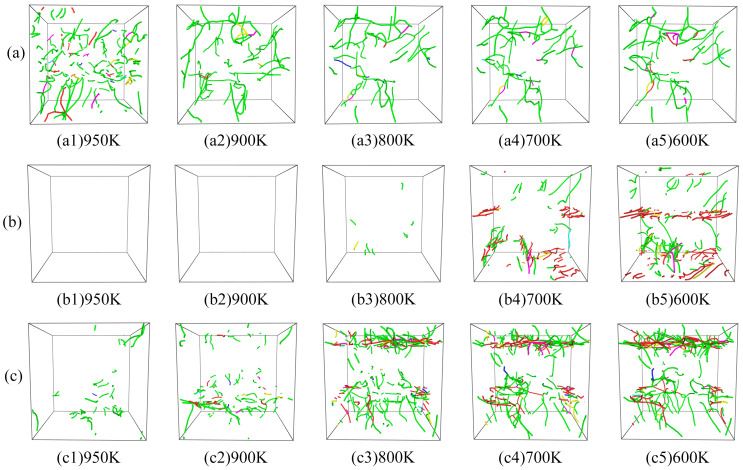
The dislocation evolution of (**a**) S1, (**b**) S2, and (**c**) S3 during rapid solidification. The green lines represent Shockley partial dislocations (Burgers vector 1/6 < 1 1 2 >); the blue lines represent perfect dislocations (Burgers vector 1/2 < 1 1 0 >); the magenta lines represent stair-rod dislocations (Burgers vector 1/6 < 1 1 0 >); the yellow lines represent Hirth dislocations (Burgers vector 1/3 < 1 0 0 >); the light blue lines represent Frank dislocations (Burgers vector 1/3 < 1 1 1 >); the red lines represent other dislocations.

**Figure 8 molecules-29-02230-f008:**
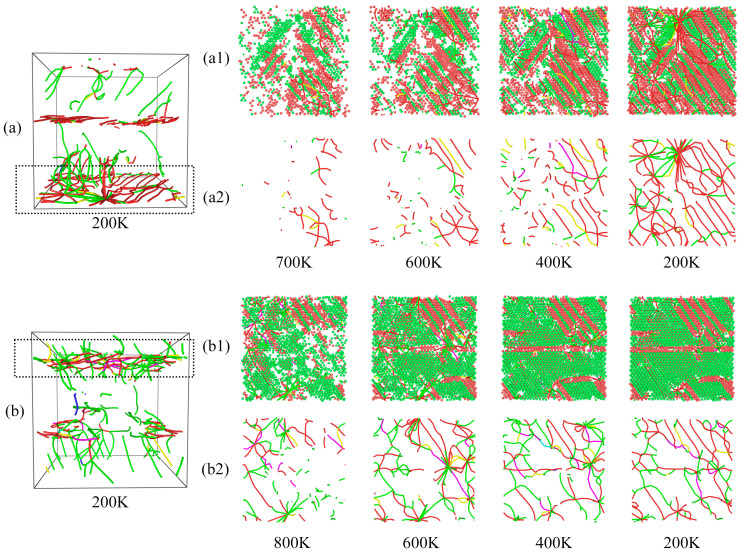
Dislocation distribution plots of (**a**) S2 and (**b**) S3 at 200 K. (**a1**,**b1**) are snapshots of the DXA of the slice containing the FCC and HCP structures, and (**a2**,**b2**) are snapshots of the slice dislocation evolution. The red atoms are HCP structures. The green lines represent Shockley partial dislocations (Burgers vector 1/6 < 1 1 2 >); the blue lines represent perfect dislocations (Burgers vector 1/2 < 1 1 0 >); the magenta lines represent stair-rod dislocations (Burgers vector 1/6 < 1 1 0 >); the yellow lines represent Hirth dislocations (Burgers vector 1/3 < 1 0 0 >); the light blue lines represent Frank dislocations (Burgers vector 1/3 < 1 1 1 >); the red lines represent other dislocations.

**Figure 9 molecules-29-02230-f009:**
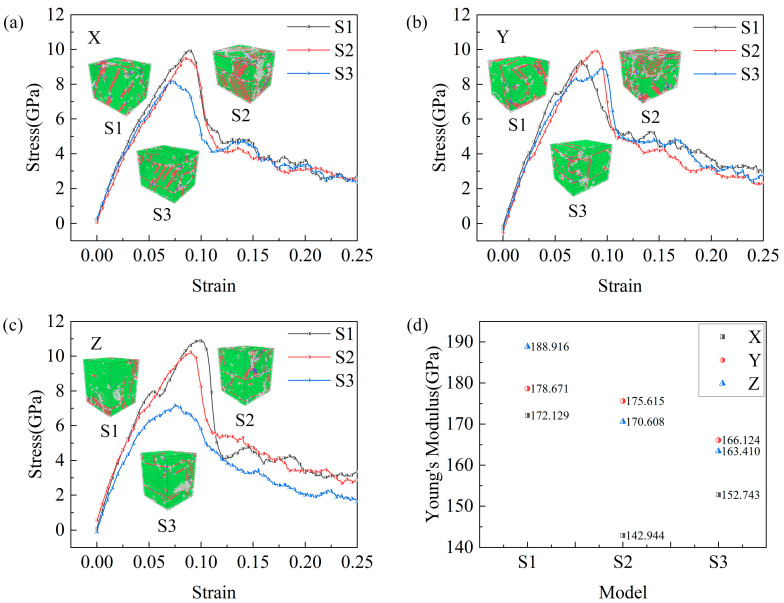
Stress–strain curves of three models stretched at 200 K for (**a**) *X*-axis, (**b**) *Y*-axis, and (**c**) *Z*-axis. (**d**) Young’s modulus.

**Figure 10 molecules-29-02230-f010:**
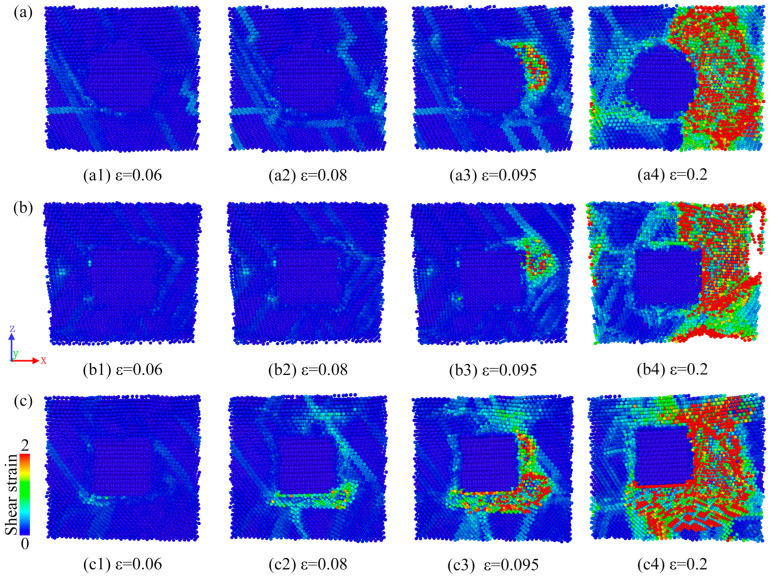
Deformation evolution of (**a**) S1, (**b**) S2, and (**c**) S3 models under *X*-axis stretching (slices).

## Data Availability

The original contributions presented in the study are included in the article, further inquiries can be directed to the corresponding authors.

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
