# Peer review of "Molecular Dynamics Simulation on Solidification Microstructure and Tensile Properties of Cu/SiC Composites"

_molecules, 2024, doi:10.3390/molecules29102230_

Round 1

Reviewer 1 Report

Comments and Suggestions for Authors

This manuscript reports molecular dynamics simulations of the metallic nanocomposites where SiC nanoparticles are embedded in a Cu matrix and quenched. The systems' crystallisation and mechanical properties are investigated for three particle shapes: spherical, cylindrical and cubic particles. The work is interesting and well-written; however, the effect of particle shape on crystallization has been studied previously and should be mentioned to show the state of the art in this area. Other minor changes must be applied to make the work suitable for publication.

-Introduction

The effect of particle shape affects many other hybrid materials, including polymers. A few significant recent MD simulation works have explained the effect of nanoparticle size, shape, and volume fraction could be mentioned. [e.g. Nanomaterials 9 (10), 1472; Nanoscale Adv., 2019,1, 4704-4721]. These molecular simulations nicely explain some experimental observations [e.g. J. Chem. Phys., 2017, 147, 020901; Polymer, 2018, 157, 111–121.]. I suggest the authors mention some of the works above to show the broad nature of the phenomena they have studied in their work on metallic nanocomposites.

Methods:

-I suggest changing the name of S3 particle from “square” to “cubic”. This is a 3D simulation, and the square is a 2D geometry.

-Please justify using the potential models in your work. Do they reproduce the experimental properties of these systems? Please provide references.

-The simulation duration for cooling is very short; 100 ps is too short. This is quenching, basically. Would changing the cooling rate affect the results?

-Why is this particle volume fraction chosen? Volume fraction, as observed in other materials [e.g. Nanomaterials 9 (10), 1472; Nanoscale Adv., 2019,1, 4704-4721], can affect the crystallinity, structure and eventual properties, which should be discussed.

Results

How do dislocations affect the SiC particles? Is the reported crystal content of FCC and HCP in Fig 4 only for the Cu matrix, or does it include structural changes in SiC as well?

-          Do the strain tests cause any change in the SiC structure?

-          Please describe how strain and stress are measured for uniaxial tensile simulations. Some guidance and previous work could be mentioned and found [https://doi.org/10.1016/j.apsusc.2020.148567] for different particle shapes [https://doi.org/10.1016/j.surfcoat.2020.125880]

Reviewer 2 Report

Comments and Suggestions for Authors

This article deals with molecular dynamics simulation on solidification microstructure and tensile properties of Cu/SiC composites. It is an interesting manuscript that contains novel and original elements and presents nice results of applying the molecular dynamics approach to describe the solidification of composite microstructure.

In general, the manuscript is well-written - with clear explanations. The figures are good and the conclusions are informative.

However, it requires an improvement concerning “References”. Positions 39 and 70 require improvements. Moreover, I suggest to include the following papers:

·       M. Fathalian, E. Postek, M.Tahani, T. Sadowski, A comprehensive study of Al2O3 mechanical behaviour using density functional theory and molecular dynamics.  Molecules. 29:1165, 2024. doi.org/  10.3390/molecules29051165

·       M. Fathalian, E. Postek, T. Sadowski.  Mechanical and electronic properties of Al(111)/6H-SiC interfaces: A DFT study. Molecules, 28(11):4345-1-19, 2023 doi.org/10.3390/molecules28114345

  • M. Tahani, E. Postek, T. Sadowski,  Diffusion and Interdiffusion Study at Al- and O-Terminated Al2O3/AlSi12 Interface Using Molecular Dynamics Simulations, Materials 2023, 16(12), 4324 https://doi.org/10.3390/ma16124324
  • M. Tahani, E. Postek, T. Sadowski,  Molecular Dynamics Study of Interdiffusion for Cubic and Hexagonal SiC/Al Interfaces, Crystals 2023, 13(1), 46 https://doi.org/10.3390/cryst13010046

To summarize, I recommend the paper for publication after small improvements.

Comments on the Quality of English Language

The English language can be improved.

Reviewer 3 Report

Comments and Suggestions for Authors

This paper investigates the solidification microstructure and tensile properties of Cu/SiC composites using molecular dynamics method. The research topic is interesting. However, the manuscript needs to be improved thoroughly before publication. The following are my detailed comments to be fully addressed.

1.     The research significance and novelty of this work failed to be identified in Abstract. There are many sentences about the research results. It is better to highlight one or two most important results in Abstract.

2.     Three shapes of SiC particles are considered. How do you maintain the same volume fraction of the reinforcements? How many C atoms and Si atoms in each particle? Do the three SiC particles have the same atom numbers?

3.     Why is the heating rate 10^14K/s, while the cooling rate is 10^11K/s, how to determine them?

4.     The uniaxial tensile simulations are conducted in NVT ensemble. How to control the pressure during the tensile process?

5.     There is no validation with experimental results or other simulation results. How to guarantee the accuracy and correctness of the MD results?

6.     S1 and S3 should be isotropic in the three directions due to the symmetry. Why they show so different stress-strain curves and Young’s modulus values?

7.     Does the Cu/SiC composite have better mechanical properties than Cu/graphene composites? More recently, Cu/graphene origami composites were proposed which have novel mechanical properties such as negative Poisson’s ratio and negative thermal expansion. The authors are suggested to add related discussion to enrich the Introduction.

Comments on the Quality of English Language

It reads fine.

Reviewer 4 Report

Comments and Suggestions for Authors

The authors have conducted classical MD simulations of melting and solidification of a composites made of Cu metal with SiC cores of different shapes, names S1, S2, and S3. The study the melting, solidification, fault distribution, partial annealing and tensile strength of each composite. The interaction potentials and simulation protocols are suitable for this type of study. There are a number of issues that need to be addressed before I can recommend the publication of this manuscript in Molecules.

-Can the authors put some dimensions on Fig. 1. Although sizes are given in the text, including them on the figure will make it easier for the reader to visualize the size of the simulation box and SiC core.

-In Fig. 3, when the authors mention the FCC and HCP phases, are they referring to the Cu component of the composite and this should be explicitly mentioned.

-The results of Fig. 3, and indeed many of the other results are interesting, but seem to be from single simulations of each initial setup. These results can also be very dependent on the particular heating / cooling regime, and the NVT / NPT ensemble chosen for the simulation. The procedure leads to strongly non-equilibrium process, the details of which will be path dependent. In particular, the rate of cooling will have a great effect on the formation of faults and the annealing of the Cu phase. In this respect, it is not known how generally applicable these results may be.

-How representative are the stoichiometries of the Cu to the SiC in the composite material in the simulations, compared with that of experimental work in this area? If there were more Cu in the system, would the effect of the SiC core diminish as the distance from the core increases? 

-The shape of the SiC cores in S1 and S3 are isotropic with respect to the x, y, and z directions, but that of S2 has a different cross section in the one plane compared to the other two. Does this effect the distribution of faults and also the calculated Young Moduli in the S1, S2, and S3 cases?

Comments on the Quality of English Language

A few minor corrections can be made at the proofreading stage. 

Round 2

Reviewer 3 Report

Comments and Suggestions for Authors

This revised manuscript has improved a lot and can be accepted for publication.

Reviewer 4 Report

Comments and Suggestions for Authors

The authors have addressed my comments and the clarity of the manuscript has improved. I can now recommend the publication of this manuscript.

Comments on the Quality of English Language

A further proofreading would help remove minor typos.